# Population Pharmacokinetics of Isavuconazole in Critical Care Patients with COVID-19-Associated Pulmonary Aspergillosis and Monte Carlo Simulations of High Off-Label Doses

**DOI:** 10.3390/jof9020211

**Published:** 2023-02-06

**Authors:** Lucas Perez, Philippe Corne, Grégoire Pasquier, Céline Konecki, Meriem Sadek, Clément Le Bihan, Kada Klouche, Olivier Mathieu, Jacques Reynes, Yoann Cazaubon

**Affiliations:** 1Department of Infectious and Tropical Diseases, Montpellier University Hospital, 34090 Montpellier, France; 2Department of Intensive Care Medicine, Lapeyronie University Hospital, 34090 Montpellier, France; 3CNRS, IRD, Montpellier University, MiVEGEC, Parasitology-Mycology Laboratory, CNR Leishmania, Montpellier University Hospital, 34090 Montpellier, France; 4Department of Medical Pharmacology, HERVI EA3801, University of Reims Champagne-Ardenne (URCA), Department of Pharmacology and Toxicology, Reims University Hospital, 51100 Reims, France; 5Department of Anesthesia and Critical Care—Cardiothoracic and Vascular Anesthesia—Regional Expert ECMO Center, Arnaud de Villeneuve University Hospital, 34090 Montpellier, France; 6Department of Anesthesia and Intensive Care Unit, Saint-Eloi University Hospital, 34090 Montpellier, France; 7PhyMedExp, INSERM, CNRS, Montpellier University, Department of Intensive Care Medicine, Lapeyronie University Hospital, 34090 Montpellier, France; 8HSM, Montpellier University, Department of Pharmacology and Toxicology, Montpellier University Hospital, 34090 Montpellier, France; 9TRANSVIHMI Unit, Montpellier University/IRD/INSERM, Department of Infectious and Tropical Diseases, Montpellier University Hospital, 34090 Montpellier, France; 10Institute Desbrest of Epidemiology and Public Health, INSERM, Montpellier University, Department of Pharmacology and Toxicology, Montpellier University Hospital, 34090 Montpellier, France

**Keywords:** isavuconazole, population pharmacokinetics, CAPA, renal replacement therapy, COVID-19, aspergillosis

## Abstract

Isavuconazole is a triazole antifungal agent recently recommended as first-line therapy for invasive pulmonary aspergillosis. With the COVID-19 pandemic, cases of COVID-19-associated pulmonary aspergillosis (CAPA) have been described with a prevalence ranging from 5 to 30%. We developed and validated a population pharmacokinetic (PKpop) model of isavuconazole plasma concentrations in intensive care unit patients with CAPA. Nonlinear mixed-effect modeling Monolix software were used for PK analysis of 65 plasma trough concentrations from 18 patients. PK parameters were best estimated with a one-compartment model. The mean of ISA plasma concentrations was 1.87 [1.29–2.25] mg/L despite prolonged loading dose (72 h for one-third) and a mean maintenance dose of 300 mg per day. Pharmacokinetics (PK) modeling showed that renal replacement therapy (RRT) was significantly associated with under exposure, explaining a part of clearance variability. The Monte Carlo simulations suggested that the recommended dosing regimen did not achieve the trough target of 2 mg/L in a timely manner (72 h). This is the first isavuconazole PKpop model developed for CAPA critical care patients underlying the need of therapeutic drug monitoring, especially for patients under RRT.

## 1. Introduction

Since the advent of the COVID-19 outbreak in December 2019, many patients have been hospitalized with viral pneumonia and respiratory failure. Nearly 5% of these patients require heavy supportive care in an intensive care unit (ICU) [1,2].

Invasive pulmonary aspergillosis is a complication described in many case series in patients with seasonal influenza hospitalized in ICUs [3,4]. SARS-CoV-2 infection (COVID-19) may also be associated with a high risk of fungal infection [5]. Several series have been published suggesting an increased risk of COVID-19-associated pulmonary aspergillosis (CAPA), with a prevalence ranging from 5% to more than 30%, resulting in increased mortality [6,7]. Mortality due to CAPA is difficult to assess in these severe patients [8]. International recommendations were published in 2020 in the Lancet Infectious Diseases in an attempt to harmonize the diagnostic criteria and management of CAPA. The expert panel recommended either voriconazole or isavuconazole as first-line treatment for possible, probable, or proven CAPA [9].

Voriconazole is a first-line treatment for invasive aspergillosis (excluding hematological malignancies). However, its use in the context of severe COVID-19 has several drawbacks regarding its pharmacokinetic characteristics: high inter-individual variability, high risk of drug–drug interactions (metabolized through CYP2C19, CYP2C9, and CYP3A4), particularly in intensive care, and non-linear pharmacokinetics at therapeutic doses [10,11]. Sub-therapeutic concentrations have been associated in the literature with a poorer prognosis [12,13], and supra-therapeutic doses with an increased risk of toxicity [14,15]. All this justifies that there are other first-line treatment options. Isavuconazole is a more recently developed azole approved for the treatment of invasive aspergillosis and mucormycosis [16,17]. Isavuconazole is characterized by high oral bioavailability (98%), high plasma protein binding, a long elimination half-life (100–130 h), and hepatic metabolism by CYP3A4 and CYP3A5 [18]. The recommended dosing regimen for both oral and intravenous formulations is a loading dose of 600 mg, administered at 200 mg every 8 h for 2 days, followed by 200 mg daily thereafter. In the SECURE trial, isavuconazole compared with voriconazole demonstrated similar efficacy and better tolerability in the treatment of invasive aspergillosis [19]. In addition, isavuconazole showed less inter- and intra-individual variability than voriconazole [20,21,22]. In a recent post hoc analysis of the SECURE trial, Desai et al. found no relationship between exposure, as represented by the area under the curve, and clinical outcome. The authors concluded that there was no clear relationship between isavuconazole exposure and clinical outcome, consequently stating that there was no evidence to support the use of TDM [23]. There is no consensus on the target concentration of isavuconazole to be reached. Some studies set it at 1 mg/L [22,24], others at 2 mg/L [25,26]. However, the EUCAST has positioned itself with thresholds of sensitivity ≤ 1 mg/L and resistance > 2 mg/L, with a clinical breakpoint set at 2 mg/L for *Aspergillus fumigatus* and *Aspergillus flavus* [27,28]. In a cohort of real-life patients, out of 283 concentrations analyzed without information on the timing of blood sampling or clinical information, the authors found a median of 2.6 mg/L that was statistically lower than the median of 3.2 mg/L found by the SECURE trial. However, since a residual level of >1 mg/L was found in 90% of patients, the authors also concluded that there was no usefulness of TDM [22].

ICU patients hospitalized for acute respiratory distress syndrome (ARDS) are likely to receive treatments such as renal replacement therapy (RRT) and extracorporeal membrane oxygenation (ECMO), and to have significant changes in their volume of distribution, especially in obese patients, making drug concentrations difficult to predict. In addition, frequent polymedication exposes patients to an increased risk of drug interactions. There is very little real-life data on isavuconazole in ICU patients, and to date there are no specific studies on patients hospitalized for CAPA. The learned societies recommend pharmacological therapeutic monitoring in patients on RRT or other extracorporeal therapies and in obese patients, stressing the importance of obtaining publications on this point [24,29]. The high prevalence and mortality associated with invasive fungal infections (IFI), particularly invasive aspergillosis, underscore the need for a proactive screening strategy in patients with severe COVID-19 and to clarify the therapeutic strategy in these patients. The aim of this retrospective study was (i) to perform a population pharmacokinetic analysis of isavuconazole in CAPA critical care patients and to identify factors influencing therapeutic underdosing; and (ii) to simulate different dosing regimens to quantify trough concentration at 72 h and 7 days (~steady-state).

## 2. Materials and Methods

### 2.1. Study Design, Subjects, Sample Collection, and Assay Method

We conducted a monocentric retrospective study in the Montpellier University Hospital, in accordance with the recommendations of good practice and the requirements of the Declaration of Helsinki. The study received approval from the Institutional Review Board under approval number IRB-MTP_2022_07_202201176. ICU patients hospitalized between March 2020 and November 2021 for ARDS on COVID-19 lung disease and with plasma isavuconazole samples were included in the study. Data were collected on an anonymized database from the patients’ computerized records. They were classified as possible/likely or proven CAPA cases according to the classification of Koehler et al. [9]. CAPA cases with an isavuconazole prescription but no sample were excluded from the analysis. The demographic, pathophysiologic, biologic (pharmacology, biochemistry, bacteriology, virology), and associated treatments data were collected after analysis of the computerized patient record and completed by review of the ICU records.

Blood samples were collected at trough during routine TDM after loading dose. The total form of isavuconazole plasma concentrations were measured by a high-performance liquid chromatography-diode array detector using liquid–liquid extraction robust against HIL interferences. This assay method was adapted from a previously published voriconazole assay method [30]. The intra- and inter-day coefficients of variation for the low and high quality controls were <10%. The lower limit of quantification was 0.5 mg/L.

### 2.2. Population Pharmacokinetic Modeling

Plasma isavuconazole concentrations were analyzed with a nonlinear mixed-effects modeling approach using the stochastic approximation maximization (SAEM) algorithm implemented in Monolix software (version 2021R2, Lixoft, Antony, France, https://lixoft.com/, accessed on 18 October 2022) combined with a Markov Chain Monte Carlo (MCMC) process [31,32]. All of the individual PK parameters were assumed to be log-normally distributed. Exponential random effects were used to describe between-subject variability (BSV). Goodness-of-fit (GOF) plots were performed using R software (version 4.2.0; http://www.r-project.org/, accessed on 18 October 2022).

#### 2.2.1. First Step—Basic Model Building

Based on the literature, one- and two-compartment models with first-order elimination were initially compared. Several error models (additive, proportional, or combined error model) were assessed for describing the residual variability (ε).

#### 2.2.2. Second Step—Covariate Analysis

Using the basic model (without covariates), the effects of twelve continuous and two categorical covariates on isavuconazole PK parameters were evaluated: age, total body weight, lean body weight, body mass index (BMI), percentage body fat, AST, ALT, total bilirubin, albumin, CKD-EPI, sequential organ failure assessment (SOFA) score, simplified acute physiology score II (SAPS II), RRT (yes/no: 1/0), and ECMO (yes/no: 1/0).

For continuous covariates, the parameter-covariate relationships were modeled as follows:$${CL}_{i}={CL}_{pop}\times {\left(\frac{{COV}_{i}}{{COV}_{median}}\right)}^{\beta }\times e^{\eta_{CL,i}},$$
where *β* is the regression coefficient to be estimated, *CL_i_* is the clearance (PK parameter) for subject *i*, *CL_pop_* is the clearance for the study population, *COV_i_* is the covariate value for subject *i*, and *COV_median_* is the median value of the covariate in the study population.

For categorical covariates, the general equation was:$${CL}_{i}={CL}_{pop}\times e^{\beta .COVi}\times e^{\eta_{CL,i}},$$
where *COV_i_* is 0 or 1 (0: non-RRT; 1: RRT).

A covariate was kept in the model if it improved the fit, reduced interpatient variability, and decreased the objective function (corresponding to −2log likelihood) and Bayesian information criterion (BIC). The statistical significance of a covariate was evaluated using the likelihood ratio test (LRT). The covariate was retained in the model if its addition reduced the LRT statistics by at least 3.84 (χ2 *p* < 0.05 for one degree of freedom). Next, during the stepwise deletion from the full model, the covariate was statistically significant if the LRT statistics were increased by 10.83 (χ2 *p* < 0.001 for one degree of freedom) or more when the covariate was deleted from the model. Finally, the Wald test (stochastic approximation) *p*-value had to be less than 0.05 to implement the covariate in the final model.

### 2.3. Internal Evaluation of the Model

Evaluation of the model was based on goodness-of-fit plots using observations versus individual and populations predictions (OBS-IPRED/PPRED), individual weighted residuals (IWRES) versus individual predictions and time, and plots of normalized prediction distribution error (NPDE) versus population predictions and time [33]. The prediction-corrected visual predictive checks (pcVPC) were performed with 1000 simulated data sets [34]. This plot showed the time course of the 10th, 50th, and 90th percentiles with 90% level of confidence around the simulated profiles and compared with the observed data [35].

### 2.4. Model Qualification Process

The final model was qualified based on the following criteria: decrease in objective function and BIC; analysis of the GOF plots (OBS-PRED, IWRES, NPDE, pcVPC); a decrease in BSV; and a low relative standard error on fixed effects (RSE < 30%) and random effects/residuals (RSE < 50%).

In the final model, the 95% confidence interval of each parameter was calculated from 500 nonparametric bootstraps based on resampling [36] using R package Rsmlx [37] and from Jackknife resampling.

The convergence assessment tool of Monolix was used for 5 estimation runs to assess the robustness of the convergence by different randomly generated initial fixed effects values as well as different seeds.

### 2.5. Monte Carlo Simulation Assessment for Maximum and Off-Label Dose Regimens

Monte Carlo simulations were performed with Simulx (mlxR: R package version 4.2.0) using the final PK model with covariate to generate 1000 isavuconazole PK virtual patients for each candidate regimen. Simulated trough concentrations were obtained for each simulated scenario between 0 and 7 days. One standard maximum and two off-label dose regimens were investigated: 3 days loading dose (LD) at 600 or 800 mg/day, followed by a maintenance dose (MD) of 200 or 400 mg/day.

## 3. Results

### 3.1. Subject Characteristics

Among the patients hospitalized for ARDS on COVID-19 pneumonia at the Montpellier University Hospital between March 2020 and November 2021, 31 were treated with antifungal agents for CAPA, including 26 with isavuconazole (Figure 1). Among them, eight patients had no isavuconazole concentration, five patients died early without sample within a median of four days after initiation of isavuconazole treatment, and three patients were lost to follow-up before the end of treatment due to transfer to another center. A total of 18 patients (14 males and four females) were included, with a median BMI of 29.2 [25.6–31.8] kg/m^2^, a median (IQR) age of 65 (56–70) years, and a median of three TDM occasions per patient. A total of 69 plasma concentrations were collected from the 18-patient cohort, but four were excluded due to uncertainty on the sampling time. Thus, 65 plasma concentrations were included in the PK analysis, provided that the treatment start date and collection time were known. The median (IQR) time to first sampling after therapy initiation was 5 (4.25–7.5) days, and the interval between initiation of isavuconazole therapy and the first plasma concentration within the therapeutic range (2–5 mg/L) was 13 (10–17.5) days (*n* = 14). The characteristics of the study population are provided in Table 1.

Six (33.3%) patients had diabetes mellitus, eight (44.4%) had hypertension, four (22.2%) had heart failure, two (11.1%) had chronic obstructive pulmonary disease, and three (16.7%) had a solid organ transplant. Among the included patients, all were on mechanical ventilation, seven (38.9%) under RRT and five (27.8%) under ECMO. For patients under RRT, three (50%) had CVVHD, one (16.7%) had CVVHDF, and two (33.3%) had SLED. For information, a comparison of isavuconazole residuals between non-dialysis patients and dialysis patients is presented in the Appendix A with a stratification of the different types of dialysis (Appendix A). All patients were treated with 6 mg of dexamethasone per day. Regarding isavuconazole treatment, all patients (*n* = 18) received intravenous isavuconazole, and four switched to *per os* during maintenance dose. Six (33.3%) patients received a prolonged loading dose (72 h instead of 48 h), and the mean maintenance dose was 264 mg/day. Bronchoalveolar lavage (BAL) culture was positive in 72.2% (13/17) of cases. The most frequently identified species was *Aspergillus fumigatus* (46.15%), followed by *Aspergillus flavus* (23%). Galactomannan was positive in BAL in 83.3% of patients, and in blood in 33.3% of patients. The host criterion was found in 27.7% of cases, the imaging criterion being difficult to interpret in the context of ARDS. Adverse effects [hepatotoxicity (*n* = 1), nausea/vomiting (*n* = 1)] probably associated with isavuconazole treatment without exceeding the toxicity threshold (5 mg/L) were reported for two (11.1%) patients. Six (33.3%) patients died during hospitalization, including four (22.2%) as a result of aspergillosis (Table 1).

### 3.2. Basic and Covariate Model Building

A total of 65 concentrations from 18 subjects were available for model development. A one-compartment model with linear elimination was identified as the optimal model to describe the PK of isavuconazole. A proportional error model (with coefficient b = 0.17) was used to describe the residual variability. RRT had a significant effect on clearance (Cl), and was the only covariate retained in the final model. It was associated with a reduction in the interindividual variability of Cl (ωCl) from 32% to 25.8%. The PK parameters of the basic (without covariate) and final (with covariate) models are summarized in Table 2. Finally, only six patients were on RRT during isavuconazole sampling, and, among them, two patients were partially on RRT during sampling. To account for the change in the RRT covariate over time (dialysis or non-dialysis periods), it was tagged as a regressor. As the Wald test *p*-value was not automatically calculated in this case, we did so externally by testing whether the shape parameter βCl was significantly different from zero (Appendix A).

### 3.3. Internal Evaluation and Validation of the Final Model

The standard error of the PK parameter estimates was less than 30% for the structural and residual error parameters, and less than 50% for random effects. The goodness-of-fit plots of the final model are presented in Figure 2 and Figure 3. Observed and predicted concentrations of isavuconazole matched well according to the plots of individual weighted residuals (IWRES) versus time after dose and versus individual predictions. The mean and variance of IWRES values were not significantly different from zero (*p* = 0.75, Student’s unpaired *t* test) and one (*p* = 0.054, Fisher variance test), respectively. Their distribution was not different from a normal distribution (*p* = 0.94, Shapiro–Wilk test). Similarly, no major systematic bias was observed for NPDE (Figure 2). The mean and variance of NPDE values were not significantly different from zero (*p* = 0.23, Student’s unpaired *t* test) and one (*p* = 0.31, Fisher variance test), respectively. Their distribution was not different from a normal distribution (*p* = 0.75, Shapiro–Wilk test). Finally, the results for 500 bootstrap replicates and Jackknife resampling are summarized in Table 2.

The pcVPC plot is presented in Figure 3, which indicated a good predictive performance of the model. Overall, the 10th, 50th, and 90th percentiles of observed isavuconazole concentrations were within the predicted 90% confidence interval of these percentiles. The mean estimates for all parameters were quite comparable to those estimates from the best covariate model. No bootstrapping runs failed. These results showed the adequacy and stability of the model. Moreover, the convergence assessment tool showed a good robustness of the convergence (data not shown).

### 3.4. Monte Carlo Simulation of Maximum/Off-Label Dosing Regimens

Figure 4 shows the evolution of simulated isavuconazole trough concentrations from 0 to 7 days for different dosing regimens. Regarding plasma concentrations, the probability of target attainment (PTA) (>2 mg/L) for a standard treatment regimen was 19% and 6% at 72 h and 18% and 2% at 7 days for non-RRT and RRT patients, respectively.

For off-label dosage, PTA for a prolonged loading dose (LD 72 h and MD 200 mg/day) was 34% and 20% at 72 h and 41% and 7% at 7 days for non-RRT and RRT patients, respectively. PTA for a prolonged loading dose and augmented maintenance dose (LD 72 h and MD 400 mg/day) was 85% and 53% at 7 days for non-RRT and RRT patients, respectively. Finally, PTA with a 72 h loading dose of 800 mg/day and a maintenance dose of 400 mg/day was 69% and 57% at 72 h and 94% and 72% at 7 days for non-RRT and RRT patients, respectively. For the highest dosing regimen, the probability of being above 5 mg/L was 1.4% and 0% at 72 h and 0.1% and 0% at 7 days for non-RRT and RRT patients, respectively.

## 4. Discussion

We studied the pharmacokinetics of intravenously administered isavuconazole in 18 hospitalized patients with CAPA using a population pharmacokinetic approach. We developed a model to reliably predict plasma isavuconazole concentrations in our population for up 21 days of treatment to capture the steady state of isavuconazole (as most available blood concentrations are collected between 4 and 21 days). We found that isavuconazole concentrations were best predicted by a one-compartment model. To our knowledge, this is the first population PK model in ICU patients with CAPA treated with isavuconazole. Our results show that RRT has a significant effect on isavuconazole clearance. The addition of RRT as a covariate improved the predictive ability of the model.

Isavuconazole has excellent oral bioavailability (98%), making the oral and intravenous forms interchangeable. Its elimination is 45.5% renal but less than 1% as unchanged form. It is highly bound to plasma proteins (albumin) and has a large volume of distribution, as expected from a highly lipophilic molecule [38]. For the other mold-active azoles, voriconazole and posaconazole, real-life data have shown the value of TDM to ensure optimal exposure and prevent the risk of failure or toxicity [39]. On the other hand, the question of the TDM of isavuconazole is not yet resolved. Some findings suggest that TDM for isavuconazole should not be considered as mandatory. However, in unstudied clinical situations and for special populations (obesity, RRT, ECMO), it should be an option to consider [40,41,42].

Uncertainties remain regarding the therapeutic range to be reached. In a post hoc study of the SECURE trial, no association was found between clinical evolution and exposure to treatment (AUC) [23]. However, other studies contradicted this conclusion and found a link between the level of antifungal exposure and the susceptibility of clinical *Aspergillus fumigatus* isolates [43] or clinical outcome [44]. A recent French study compared plasma trough concentrations of isavuconazole with the concentration found at the zone of fungal growth inhibition in order to specify the clinical therapeutic threshold [45]. A concentration of 2 mg/L conferred 52% of the maximum in vitro antifungal activity on *Aspergillus fumigatus*, similar to that observed with posaconazole at therapeutic dose. In addition, the authors emphasized the negative impact of plasma on the antifungal activity, which may overestimate the efficacy found in vitro for isavuconazole in this study, as similar results were found in studies on posaconazole [46]. Furthermore, isavuconazole appears to have significant inter-individual variability [23,47]. Microbiologically, the EUCAST threshold in 01/2022 was set at 2 mg/L for isavuconazole [28]. A plasma concentration ≥ 2 mg/L seems to provide better exposure for all patients [48]. The study by Furfaro et al. [20] defined a toxicity threshold of 5 mg/L, in particular to describe the occurrence of gastrointestinal adverse drug reactions. Although there is still uncertainty about the therapeutic range of isavuconazole, many consider that the therapeutic range is between 2 and 5 mg/L [21].

Real-life data concerning isavuconazole concentrations are scarce and involve very heterogeneous study populations. In a recent study, 283 samples were analyzed, but details of the clinical context were not available. In these patients, the median Cmin found was 2.6 mg/L without specifying the time period evaluated [22]. In Zurl et al.’s study concerning 32 patients with onco-hematological diseases, the median Cmin was 3.05 mg/L [24]. Data from three phase 3 clinical trials (*n* = 2458 concentrations of 551 patients) showed results comparable to the previously cited study (median Cmin was 3.02 mg/L at D35) [19]. In our study, the median Cmin was 1.87 mg/L (1.29–2.25) over the entire follow-up period, lower than those reported in clinical trials or in previous real-life studies. At D35, the median trough concentrations in our CAPA cohort were 2.4 mg/L, close to Zurl et al.’s study, but dosing regimens were on average higher in our cohort (LD 60 h and MD 264 mg/d) than in Zurl et al.’s cohort (LD 48 h and MD 200 mg/day) [29]. For patients who required RRT during isavuconazole treatment, the median Cmin at D35 was even lower (1.87 mg/L).

Our study showed that the behavior of isavuconazole is different from that described by previous studies. In our cohort, the mean clearance was estimated at 6.24 L/h for patients under RRT and 3.98 L/h for non-RRT patients. The molecular weight of isavuconazole is less than 500 daltons, but its very lipophilic character (logP = 3.5), its high protein binding (predominantly albumin, >99%), and its high volume of distribution >1 L/kg do not allow us to say that isavuconazole is a good candidate for conventional hemodialysis. However, in critically ill circumstances with severe hypoalbuminemia, the binding of isavuconazole to other plasma components, and pH abnormalities, all these biological variations make it difficult to predict the dialyzability of isavuconazole. In a study of neutropenic ICU patients, isavuconazole plasma concentrations were significantly lower after SLED therapy (3.36 vs. 5.73 mg/L; *p* < 0.001), representing a 42% decrease in drug levels [29]. In our study, patients with lower plasma concentrations were also SLED patients. The characteristics of the dialysis could be a factor in lowering plasma concentrations, but our small number of patients did not allow us to determine the effect of dialysis mode, filter type, blood flow rate, and effluent flow rate, but to consider only whether the patient was on dialysis or not (Appendix A). Another possible explanation is adsorption on the dialysis membrane [49,50]. Biagi et al. concluded the opposite result, in a study performed on a cohort of transplant patients receiving isavuconazole as prophylaxis. They found Cmin concentrations at 1.76 ± 0.76 mg/L, with the calculated transmembrane clearance representing 0.7% of the total clearance of isavuconazole [51].

Several factors could explain the observed discrepancies against the previous literature. First, isavuconazole is a substrate of cytochrome P450 3A4/3A5 and may be subject to drug–drug interactions. In our cohort, concomitant administration of cytochrome P450 3A4 inhibitors was found for some patients: three patients received amiodarone (moderate inhibitor), one received esomeprazole (moderate inhibitor), and one received esomeprazole and amiodarone. For these patients exposed to moderate inhibitors, the concentrations were not higher compared to the median. This did not appear to have a significant impact on isavuconazole metabolism (data not shown). In contrast, all patients were exposed to dexamethasone at 6 mg per day during their isavuconazole treatment. Although it is a moderate inducer of CYP3A4, the high doses received could potentially have had an impact on increasing clearance [52,53]. Furthermore, the observed increase clearance could be related to the patients’ CYP3A-genotype. Indeed, whereas 85% of Europeans (*n* = 15 in our cohort) and 77% of Middle Eastern people (*n* = 3) show a poor metabolizer status for CYP3A5, 13%/21% of Europeans/Middle Eastern people carry the CYP3A5*1/*3 diplotype, conferring an intermediate metabolizer status and thus quicker metabolism [54]. However, these frequencies are too low to explain the increased clearance in our study population. Finally, for the volume of distribution, we found a mean value of 850 L, very close to that reported by Cojutti et al. (835 L) [55], but much higher than those reported by Desai et al. [56], Kovanda et al. [57], or Wu et al. [58]. This elevated volume of distribution could be explained by an elevated BMI in our cohort (29.2 kg/m^2^). Moreover, hypoalbuminemia was observed in our patients, which could also have played a role in the increase of the volume of distribution. But the small number of patients in our cohort with all hypoalbuminemic patients does not allow sufficient statistical power to demonstrate the effect of this covariate on this PK parameter.

The use of ECMO in the ICU has been described as possibly playing a role in pharmacokinetic changes in some patients and, particularly with lipophilic and highly protein-bound therapies, that may be captured by the ECMO system [59]. In the study by Kriegl et al., no difference was found between pre- and post-membrane concentrations, therefore excluding adsorption on the membrane [60]. The data in the literature are limited concerning isavuconazole and ECMO. In the study by Zurl et al., isavuconazole concentrations were significantly lower in patients undergoing RRT associated for some with ECMO (median (IQR): 0.88 (0.71–1.21) mg/L) [24,61]. In our cohort we could not show that ECMO was related to a change in the pharmacokinetic profile and under-exposure. Patients in our cohort likely had multiple confounding factors and our enrollment was limited (ECMO, *n* = 5).

From the final model, Monte Carlo simulations were performed to determine the optimal dosing regimen in our population to achieve a target concentration of 2 mg/L at 72 h and 7 days. The regimen with a loading dose of 800 mg/day over 72 h and a maintenance dose of 400 mg achieved the target in 69% and 57% of cases at 72 h, and 94% and 72% at 7 days for non-RRT and RRT patients, respectively. Our data suggest that, in these ICU patients for whom ensuring effective exposure as quickly as possible is a major issue, a dosing regimen higher than that recommended (LD 48 h 600 mg/day and MD 200 mg/day) could be proposed. 

Our study had some limitations. First, the main limit is the small number of patients who could be included. Second, the modeling was performed using data from daily practice, so the sampling design was not adapted to validating a two-compartment model as in most of the carried out studies [55,56,57]. Moreover, we could not find any impact of other covariates described in the literature as being able to influence isavuconazole concentrations. In a recent study in the ICU, in which CAPA was excluded, the median Cmin was 1.74 mg/L (0.24–4.96) in 82 trough concentrations and 31.7% of samples were below 1 mg/L. BMI and SOFA score (>12) were associated with lower plasma concentrations [62]. Furthermore, in the study by Wu et al., peripheral volume of distribution was positively correlated with BMI and clearance was higher in women [58]. This could not be demonstrated in our cohort due to a certain homogeneity of BMI in our cohort and the number of women too low compared with the number of men. Finally, it was not possible to statistically study the link between exposure and outcomes (mortality or adverse drug reactions), as the number of patients in each group was small.

In our CAPA cohort, we observed that COVID-19 generated, for patients admitted to the ICU, the need for many very heavy and cumulative supportive care: ECMO, RRT, 100% on mechanical ventilation, 100% on high-dose dexamethasone, and 100% on broad-spectrum antibiotic therapy combined with a median BMI of 30 and IFI. One could afford to use the term “supportive care and pathophysiology package” with respect to COVID-19 (69). This “package” could explain the difference in magnitude of the PK parameters compared with previous studies [55,56,57,58].

## 5. Conclusions

We reported the first isavuconazole pharmacokinetic model in real-life ICU patients with CAPA. Simulation based on this model suggests that the recommended dosage achieves the therapeutic target ≥2 mg/L at 72 h for an extremely low percentage of non-RRT CAPA patients (19%) and even lower for the percentage of dialyzed patients (6%). The Monte Carlo simulations showed that off-label dosage regimens are required. In this particular population where cumulative supportive care seemed to have an impact on the PK of isavuconazole, this confirms the need for TDM in CAPA patients, especially with those under RRT. For CAPA patients (RRT or non-RRT), in order to secure and optimize therapeutic management, 800 mg/day during 3 days followed by 400 mg/day seems necessary. At this stage, two approaches are possible: either the empirical approach consisting of performing TDM at the end of the loading dose and readjusting the maintenance dose later, or the rational approach consisting of taking a blood sample at H48 in order to adjust the maintenance dosage at H72 using the PK model. In the event that the patient is dialyzed several days after the initiation of isavuconazole, TDM should be performed to assess the impact of dialysis on the patient’s exposure. In case of underdosing, a re-loading dose should be necessary.

## Figures and Tables

**Figure 1 jof-09-00211-f001:**
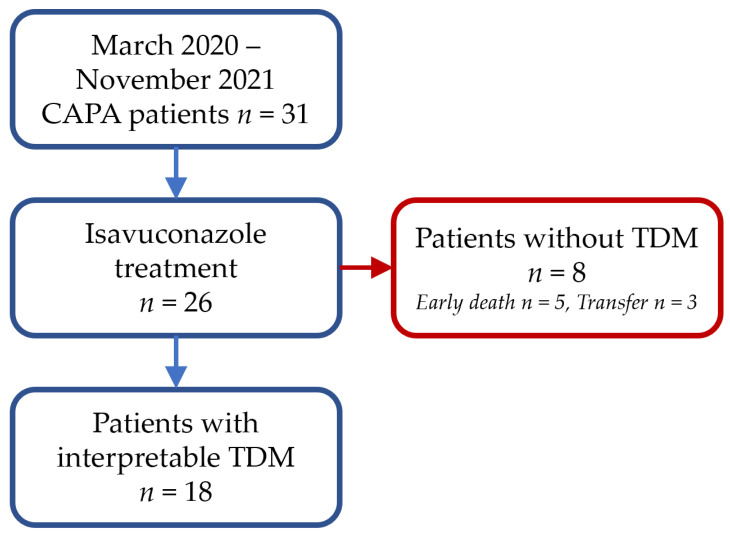
Study flow chart.

**Figure 2 jof-09-00211-f002:**
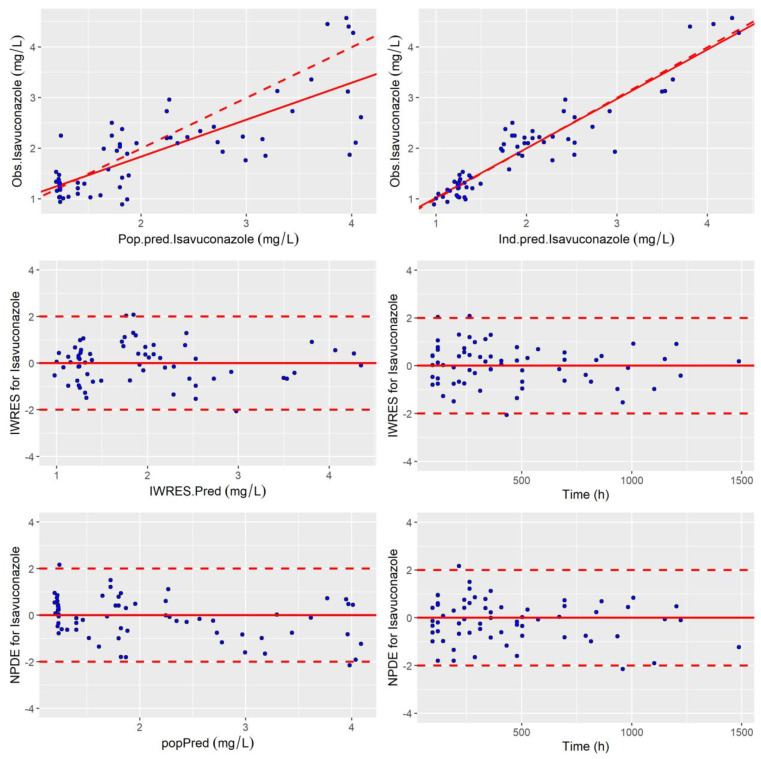
Goodness of fit plots: Obs vs. Pred, IWRES and NDPE. (**top-left**) Population predicted isavuconazole concentrations versus observed concentrations (Rho Spearman = 0.71). (**top-right**) Individual predicted isavuconazole concentrations versus observed concentrations (Rho Spearman = 0.89). The linear regression line of fit is shown by the continuous line and the line of identity xy is shown by the broken line. Individual weighted residual (IWRES) as a function of individual predictions (**middle-left**) and time (**middle-right**). Normalized prediction distribution error (NPDE) (**bottom-left**) as a function of population-predicted (popPred) concentrations of isavuconazole and (**bottom-right**) as a function of time.

**Figure 3 jof-09-00211-f003:**
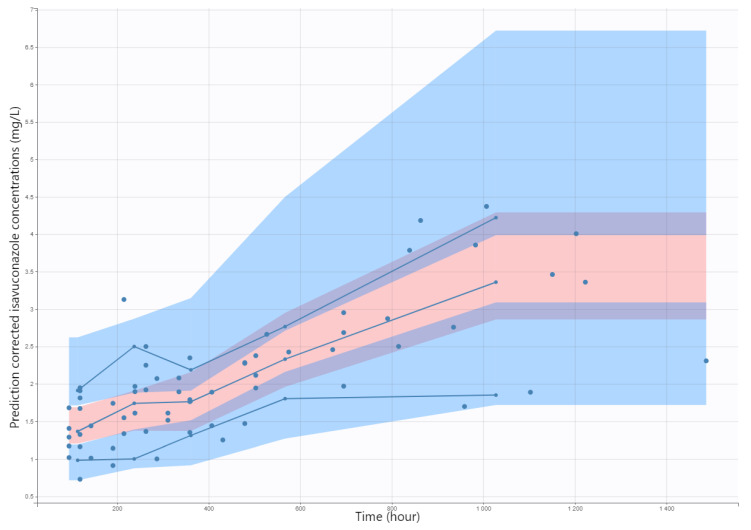
Prediction-corrected visual predictive check (pcVPC) for isavuconazole (mg/L). Blue dots are observed isavuconazole (mg/L). Solid blue lines represent the median, 10th, and 90th percentile of the observed values and shaded areas represent the spread of 90% prediction intervals calculated from simulations.

**Figure 4 jof-09-00211-f004:**
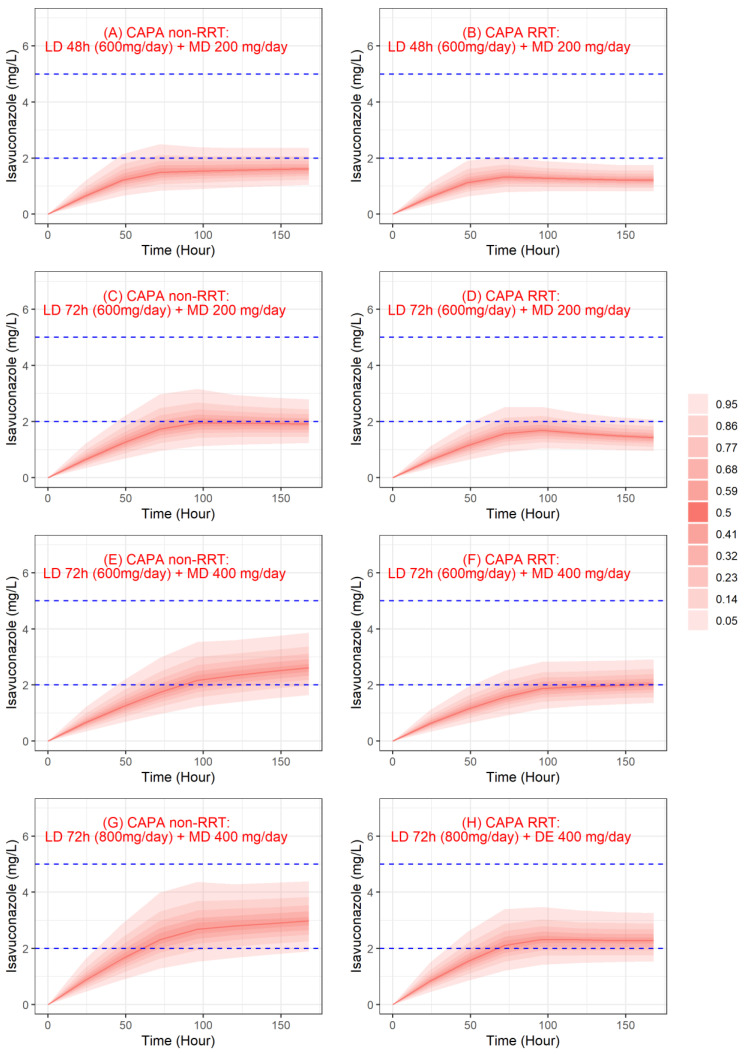
Simulated trough concentrations profile of isavuconazole. Red lines and red shaded areas, respectively, represent the median and the confidence interval of simulated concentrations, and blue dashed lines represent the therapeutic range for isavuconazole concentrations. CAPA: COVID-19-associated pulmonary aspergillosis; LD: loading dose; MD: maintenance dose; RTT: renal replacement therapy.

**Table 1 jof-09-00211-t001:** Characteristics of CAPA patients on the whole isavuconazole TDM.

Characteristics	Patients (*n* = 18)
**Clinical features**	
Sex, M/F	14/4
Age (years), Median (IQR)	65 (56–70)
Weight (kg)/BMI (kg/m^2^), Median (IQR)	85 (79–92)/29.2 (25.6–31.8)
Underlying disease	
Diabetes mellitus, *n* (%)	6 (33.3)
Hypertension, *n* (%)	8 (44.4)
Heart failure, *n* (%)	4 (22.2)
COPD, *n* (%)	2 (11.1)
Solid organ transplant, *n* (%)	3 (16.7)
SOFA score at admission, Median (IQR)	5.5 (3.3–8.8)
Mechanical ventilation, *n* (%)	18 (100)
RRT/RTT during isavuconazole therapy, *n* (%)	7 (38.9)/6 (33.3%)
CVVHD/CVVHDF/SLED, *n* (%)	3 (50%)/1 (16.7%)/2 (33.3%)
ECMO, *n* (%)	5 (27.8)
**Biological parameters**	
C reactive protein (mg/L), Median (IQR)	101 (53–159)
White blood cells (G/L), Median (IQR)	13.1 (9.5–14.6)
Platelets (G/L), Median (IQR)	283 (174–362)
Hemoglobin (g/dL), Median (IQR)	9.4 (9–10.9)
Hemotocrit (%), Median (IQR)	28.2 (26.7–32)
Albumin (g/L), Median (IQR)	28.5 (26–31.8)
eGFR (mL/min/1.73 m^2^) estimated by CKD-EPI, Median (IQR)	89.3 (60.5–98.9)
ASAT (U/L)/ALAT (U/L), Median (IQR)	34 (25.8–51.8)/44.8 (30.3–70.3)
GGT (U/L), Median (IQR)	198.3 (64–435)
Conjugated/total bilirubin (μmol/L), Median (IQR)	5 (4–6.4)/7.8 (6–10.8)
Mycological diagnosis	
*Aspergillus* galactomannan antigen in BAL, *n* (%)	15 (83.3)
BAL positive culture	13 (72.2)
*Aspergillus fumigatus*/*Aspergillus flavus*/others, *n* (%)	6 (46.2)/3 (23)/4 (30.8)
**Treatments**	
Concomitant cytochrome P450 inhibitor, *n* (%)	3 (16)
Concomitant cytochrome P450 inducer: dexamethasone 6 mg per day, *n* (%)	18 (100)
Isavuconazole 72 h LD, *n* (%)/MD (mg/day), Mean ± SD	6 (33)/264 ± 79
Intravenous route/per os switch during MD, *n* (%)	18 (100)/4 (22.2)
Time after last dose (h), Mean ± SD/Median (IQR)	18.6 ± 5.9/24 (12–24)
Isavuconazole trough concentration, Median (IQR)	1.87 (1.29–2.25)
Plasma concentrations: 2–5 mg/L, *n* (%)	28 (43)
Plasma concentrations < 2 mg/L, *n* (%)	37 (57)
Plasma concentrations > 5 mg/L, *n* (%)	0 (0)

BAL: bronchoalveolar lavage; BMI: body mass index; COPD: chronic obstructive pulmonary disease; CVVHD/F: continuous venovenous hemodialysis/filtration; GGT: gamma-glutamyl-transferase; IQR: interquartile range; LD: loading dose, MD: maintenance dose; RRT: renal replacement therapy; SOFA: sequential organ failure assessment; SLED: sustained low-efficiency dialysis.

**Table 2 jof-09-00211-t002:** Population pharmacokinetics of isavuconazole.

	Basic Model	Final Model			
	Value (RSE %)	Value (RSE %)	Shrinkage (%)	Bootstrap Mean (95% CI)	Jackknife (95% CI)
**Fixed effect**					
Cl (L.h^−1^)	4.75 (11.5)	3.98 (9.9)	–	3.97 (3.11–5.02)	4.04 (3.76–4.32)
βCl (log L.h^−1^)-RRT	–	0.45 (22.6) *	–	0.44 (0.15–0.73)	0.43 (0.33–0.51)
V (L)	838 (15)	850 (13.5)	–	837 (572–1102)	817 (745–889)
**Random effect**					
ωCl (%)	32 (25.9)	25.8 (28.8)	2.9	20.6 (4.2–37)	23.2 (18.4–28)
ωV (%)	38 (49)	39.9 (33.2)	5.1	37.4 (12–63)	42.5 (35–50)
**residual**					
b (proportional)	0.2 (12.9)	0.17 (12.2)	–	0.17 (0.11–0.23)	0.17 (0.15–0.18)
BIC	105.8	91.7	–	–	–

Abbreviations are as follows: BIC, Bayesian information criterion; RRT, renal replacement treatment; RSE, relative standard errors. * Wald test *p*-value: 6.8 × 10^−6^. Final covariate model: ${Cl}_{i}=3.98\times e^{\left(0.45\times RRT\right)}$.

## Data Availability

Due to ethical, legal, or privacy concerns, and in accordance with the consent provided by participants, individual data cannot be shared.

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
