# Peer review of "Population Pharmacokinetics of Isavuconazole in Critical Care Patients with COVID-19-Associated Pulmonary Aspergillosis and Monte Carlo Simulations of High Off-Label Doses"

_jof, 2023, doi:10.3390/jof9020211_

Round 1
Reviewer 1 Report
This is a population analysis of sparse, clinically obtained blood concentrations of ISAV in critically ill adults with COVID associated pulmonary aspergillosis. In general it is easy to follow, but the manuscript could use another pass by a native English speaker to correct several minor idiomatic flaws.
I do have some suggestions and queries.
- In Table 1, please include the time after dose distribution under the Treatments section.
- In the results, please explicitly state that no other covariate except RRT was included in the model.
- I don't understand the profiles in Figure 3. Are they trough concentrations joined by line segments? Even a drug with as long a half-life as isavuconazole would not increase gradually like the profiles in the figure with intermittent dosing. They look like profiles associated with continuous infusions. Please clarify what points are actually plotted/generated in the simulation.
- Please include in the discussion how your parameter estimates compare to other models/populations. There's a hint that clearance is higher when discussing dexamethasone, but nothing explicit. We are left wondering if the observed lower concentrations are due to increased Vd or increased CL. Please suggest reasons for your findings.
- It appears that the authors are advocating for the 72h loading with 800 mg/day followed by 400 mg/day. There needs to be a corresponding PTA for trough >5 as an idea of what toxicity might occur with such high doses.
- Please make a recommendation on how the dose should be adjusted for RRT. Otherwise this analysis is not very practical.
- Can the authors evaluate outcomes (e.g. attributable mortality or hepatotoxicity) vs. exposure in their cohort?
- Figure S2 needs better explanation or should be deleted. As it is now, it contributes nothing to the manuscript.
Author Response
This is a population analysis of sparse, clinically obtained blood concentrations of ISAV in critically ill adults with COVID associated pulmonary aspergillosis. In general it is easy to follow, but the manuscript could use another pass by a native English speaker to correct several minor idiomatic flaws.
As requested, the manuscript was carefully read by a native English. I hope this meets your expectations.
I do have some suggestions and queries.
- In Table 1, please include the time after dose distribution under the Treatments section.
We included time after last dose in the Treatment section (see Table 1.):
|
Time after last dose (h), Mean+/-SD |
18.6 +/- 5.9 |
- In the results, please explicitly state that no other covariate except RRT was included in the model.
As requested, we did it (line 230-231):
The best full covariate model only included effect of RRT on clearance (Cl).
I don't understand the profiles in Figure 3. Are they trough concentrations joined by line segments? Even a drug with as long a half-life as isavuconazole would not increase gradually like the profiles in the figure with intermittent dosing. They look like profiles associated with continuous infusions. Please clarify what points are actually plotted/generated in the simulation.
Yes, they are as only trough concentrations to build simulation plots. Indeed, the model is able to predict well the residual so we just focus on it. We already had specified in the method section that the simulated concentrations were troughs. We added a precision in the title of the figure 3 and in the results section (line 285):
“Figure 3. shows the evolution of simulated isavuconazole trough concentrations from 0 to 7 days for different dosing regimens.”
- Please include in the discussion how your parameter estimates compare to other models/populations. There's a hint that clearance is higher when discussing dexamethasone, but nothing explicit. We are left wondering if the observed lower concentrations are due to increased Vd or increased CL. Please suggest reasons for your findings.
Yes, I am agree with you. This paragraph simplifies the potential causes of parameter variation. So we detailed this in the following paragraph (line 352-383):
This study showed that the behavior of isavuconazole is very different from that described by previous studies. In our cohort, the mean clearance was estimated at 6.24 L/h for patients under RRT and 3.98 L/h for non-RRT patients. In a study of neutropenic ICU patients with indication for RRT, isavuconazole concentrations were significantly lower after RRT (3.36 vs. 5.73 mg/L; P < 0.001), representing a 42% decrease in drug levels [29]. Possible explanations are the type of RRT, the strong protein binding of isavuconazole, and adsorption on the dialysis membrane [48,49]. Biagi et al. concluded the opposite result, in a study performed on a cohort of transplant patients receiving isavuconazole as prophylaxis. They found Cmin concentrations at 1.76 +/- 0.76 mg/L, with the calculated transmembrane clearance representing 0.7% of the total clearance of isavuconazole [50]. Several factors could explain the observed discrepancies against previous literature. First, isavuconazole is a substrate of cytochrome P450 3A4/3A5 and may be subject to drug-drug interactions. In our cohort, concomitant administration of cytochrome P450 3A4 inhibitors was found for some patients: three patients received amiodarone (moderate inhibitor), one received esomeprazole (moderate inhibitor), and one received esomeprazole and amiodarone. For these patients exposed to moderate inhibitors, the concentrations were not higher compared to the median. This did not appear to have a significant impact on isavuconazole metabolism (data not shown). In contrast, all patients were exposed to dexamethasone at 6 mg per day during their isavuconazole treatment. Although it is a moderate inducer of CYP3A4, the high doses received could potentially have had an impact on increasing clearance [51,52]. Furthermore, the observed increase clearance could be related to the patients’ CYP3A-genotype. Indeed, whereas 85% of Europeans (n = 15 in our cohort) and 77% of Near Easterns (n = 3) show a poor metabolizer status for CYP3A5, 13%/21% of Europeans/Near Easterns carry the CYP3A5*1/*3 diplotype, conferring an intermediate metabolizer status and thus quicker metabolism [53]. However, these frequencies are too low to explain the increases clearance in our study population. Finally, for the volume of distribution, we found a mean value of 850 L, very close to that reported by Cojutti et al. (835 L) [54], but much higher than those reported by Desai et al. [55], Kovanda et al. [56], or Wu et al. [57]. This elevated volume of distribution could be explained by an elevated BMI in our cohort (29.2 kg/m²). Moreover, hypoalbuminemia was observed in our patients, which could also have played a role in the increase of the volume of distribution.
- It appears that the authors are advocating for the 72h loading with 800 mg/day followed by 400 mg/day. There needs to be a corresponding PTA for trough >5 as an idea of what toxicity might occur with such high doses.
As requested, we added PTA for trough > 5 mg/L in the section results (line 295-297):
For the highest dosing regimen, the probability of being above 5 mg/L was 1.4% and 0% at 72h and 0.1% and 0% at 7 days for non-RRT and RRT patients, respectively.
- Please make a recommendation on how the dose should be adjusted for RRT. Otherwise this analysis is not very practical.
We did it in the Conclusions section (line 432-439):
For the latter, in order to secure and optimize therapeutic management, 800 mg/day during 3 days followed by 400 mg/day seems necessary. At this stage, two approaches are possible: either the empirical approach consisting of performing TDM at the end of the loading dose and readjusting the maintenance dose later, or the rational approach consisting of taking a blood sample at H48 in order to adjust the maintenance dosage at H72 using the PK model. In the event that the patient is dialyzed several days after the initiation of isavuconazole, TDM should be performed to assess the impact of dialysis on the patient's exposure.
- Can the authors evaluate outcomes (e.g. attributable mortality or hepatotoxicity) vs. exposure in their cohort?
We did not but we added the following paragraph (line 414-416):
Finally, it was not possible to statistically study the link between exposure and outcomes (mortality or adverse drug reactions), as the number of patients in each group was small.
$$ Just as a supplement, but there is no point in concluding on anything as the number of values is too low to assess a potential link or not:
- Figure S2 needs better explanation or should be deleted. As it is now, it contributes nothing to the manuscript.
As expressed in the results section, the convergence assessment tool show a good robustness of the convergence, figure s2 has been deleted as the contribution is negligible.
Reviewer 2 Report
This paper reported Population pharmacokinetics of isavuconazole in critical care patients with COVID-19-associated pulmonary aspergillosis and Monte Carlo simulations of high off-label doses. The manuscript did achieve a high enough priority score as it has the novelty. They reported the first isavuconazole pharmacokinetic model in real-life ICU patients with CAPA. Simulation based on this model suggests that the recommended dosage achieves the therapeutic target ≥ 2 mg/L at 72h for an extremely low percentage of non-RRT CAPA patients (19%) and even lower for dialyzed patients (6%). And, there were some flaws or uncertainties in this research in terms of my knowledge, so it should be revised before publish in my opinion.
1. As reported by authors, Voriconazole is a first-line treatment for invasive aspergillosis, why authors research isavuconazole?
2. Isavuconazole is characterized by hepatic metabolism by CYP3A4 and CYP3A5. In this pharmacokinetic model, what about the liver function affect Isavuconazole metabolism? Even the gene polymorphism of CYP3A4 and CYP3A5.
Author Response
This paper reported Population pharmacokinetics of isavuconazole in critical care patients with COVID-19-associated pulmonary aspergillosis and Monte Carlo simulations of high off-label doses. The manuscript did achieve a high enough priority score as it has the novelty. They reported the first isavuconazole pharmacokinetic model in real-life ICU patients with CAPA. Simulation based on this model suggests that the recommended dosage achieves the therapeutic target ≥ 2 mg/L at 72h for an extremely low percentage of non-RRT CAPA patients (19%) and even lower for dialyzed patients (6%). And, there were some flaws or uncertainties in this research in terms of my knowledge, so it should be revised before publish in my opinion.
- As reported by authors, Voriconazole is a first-line treatment for invasive aspergillosis, why authors research isavuconazole?
In our university hospital, Isavuconazole has been chosen as the first-line treatment based on international recommendations.
In order to clarify this, we have made some adjustments in the introduction section (line 53-64):
International recommendations were published in 2020 in the Lancet Infectious Diseases in an attempt to harmonize the diagnostic criteria and management of CAPA. The expert panel recommended either voriconazole or isavuconazole as first-line treatment for possible, probable, or proven CAPA [9].
Voriconazole is a first-line treatment for invasive aspergillosis (excluding hematological malignancies). However, its use in the context of severe COVID-19 has several drawbacks regarding its pharmacokinetic characteristics: high inter-individual variability, high risk of drug-drug interactions (metabolized through CYP2C19, CYP2C9, and CYP3A4), particularly in intensive care, and non-linear pharmacokinetics at therapeutic doses [10,11]. Sub-therapeutic concentrations have been associated in the literature with a poorer prognosis [12,13] and supra-therapeutic doses with an increased risk of toxicity [14,15]. All this justifies that there are other first-line treatment options.
- Isavuconazole is characterized by hepatic metabolism by CYP3A4 and CYP3A5. In this pharmacokinetic model, what about the liver function affect Isavuconazole metabolism?
In the PK analysis, biological data that may reflect liver function (ASAT, ALAT, GGT, BILI) does not appear to have an effect on either Vd or clearance. If this were the case, the low number of patients resulting in a lack of power does not allow for statistical objectivity.
Even the gene polymorphism of CYP3A4 and CYP3A5.
We added a paragraph concerning polymorphism in discussion section (line: 372-377):
Furthermore, the observed increase clearance could be related to the patients’ CYP3A-genotype. Indeed, whereas 85% of Europeans (n = 15 in our cohort) and 77% of Near Easterns (n = 3) show a poor metabolizer status for CYP3A5, 13%/21% of Europeans/Near Easterns carry the CYP3A5*1/*3 diplotype, conferring an intermediate metabolizer status and thus quicker metabolism [53]. However, these frequencies are too low to explain the increases clearance in our study population

Reviewer 3 Report
Reviewer’s comments
Article: Population pharmacokinetics of isavuconazole in critical care patients with COVID-19-associated pulmonary and Monte Carlo simulations of high off-label doses
The objectives of this work are aim to perform PPK analysis of isavuconazole in CAPA critical care patients, to identify factors influencing underdosing and dosing simulation. One-compartment model was achieved and RRT causes the lower drug level requiring higher dosage than approval dose. The results were interesting and highlighted the usefulness of isavuconazole TDM in these population.
Major comments
1. For blood sampling time, most of samples were collected at which time point between dosing interval?
2. Please give the reference of drug assay method and the total form or unbound form of isavuconazole level was measured.
3. For model qualification by bootstrapping, Why did you choose this method? Because your samples were too small, another method such as Jackknife is more reasonable ?
4. For Monte Carlo simulation, why did you decide to simulate the drug level on Day7?
5. How long of RRT period of most patients? And RRT was initiated before or after starting treatment?, and these simulated regimens are made for the patients on RRT who are subsequently infected with CAPA? If COVID-19 patients were infected CAPA and already started isavuconazole before RRT, can you recommend using your simulation to guide the dosing? Requiring Re-loading?
6. Line212: It should be better not to use “therapeutic target 2-5 mg/L”, because now we don’t know the therapeutic range of isavuconazole. You should change to another wording.
7. Any covariate affecting on the volume of distribution? Such as serum albumin at each time point, because most of patients were hypoalbuminemia and isavuconazole is highly bound to plasma protein.
8. You should state that for non-RRT patients, there was not any covariate affecting on isavuconazole PK parameters. Because non-RRT patients were the most of all patients.
9. Discussion section, Line323 ”We developed a model to reliably predict plasma isavuconazole concentrations in our population for up to 50 days of treatment to capture the steady state of isavuconazole.” I knew that you followed the patients up to 50 days of treatment, but I’m not sure that you can use this sentence to claim that your model is reliably predicting isavuconazole for up to 50 days of treatment. Your model could predict well during the first two weeks of treatment according to your most available blood concentration.
10. It should be better if you can discuss more about the physicochemical properties of isavuconazole and the mechanism of each RRT mode to remove isavuconazole, such as isavuconazole MW around 437 dalton can be removed by CVVHD efficiently to relate to your PPK model result, to highlight this concern to the readers.
Minor comments
1. Line77, “In a recent post-hoc analysis of the SECURE trial, Desai et al. found no relationship between exposure as represented by the area under the MIC curve and clinical outcome.”
Question: I’m not sure that these bold letters are correct? You mean AUC/MIC? Or area under concentration time curve over MIC.
2. In Table 1, I’m not familiar with the unit of white blood cells and platelet count, G/L = gram/litre?.
Author Response
Dear Referee,
We want to thank you for your helpful comments improving the quality of the manuscript.
We have made some changes in the revised manuscript in accordance with your suggestions.
Article: Population pharmacokinetics of isavuconazole in critical care patients with COVID-19-associated pulmonary and Monte Carlo simulations of high off-label doses
The objectives of this work are aim to perform PPK analysis of isavuconazole in CAPA critical care patients, to identify factors influencing underdosing and dosing simulation. One-compartment model was achieved and RRT causes the lower drug level requiring higher dosage than approval dose. The results were interesting and highlighted the usefulness of isavuconazole TDM in these population.
Major comments
- For blood sampling time, most of samples were collected at which time point between dosing interval?
All of the samples were collected at trough, we put a mean time after last dose in table 1, we also added median [IDR]. Most troughs are at H24 (n = 41).
We precised it in the M & M section that blood samplings correspond to troughs (line 117):
Blood samples were collected at trough during routine TDM after loading dose
- Please give the reference of drug assay method and the total form or unbound form of isavuconazole level was measured.
It is an in-house assay method adapted from a published voriconazole assay method. Only the total form of isavuconazole was measured.
We precised it in the M &M section (line 117-120)
Total form of isavuconazole plasma concentrations were measured by high-performance liquid chromatography-diode array detector using liquid-liquid extraction robust against HIL interferences. This assay method was adapted from a previously published voriconazole assay method.
- For model qualification by bootstrapping, Why did you choose this method? Because your samples were too small, another method such as Jackknife is more reasonable ?
We chose non parametric bootstrapping because it is the most robust resampling method. However, on a small sample with sparse data, it can lead to overestimated uncertainty of parameters in non-linear mixed effect models with heteroscedactic error. We therefore followed your recommendations by doing a Jackknife which is more suitable for internal validation on a small sample. The average values of the parameters are comparable to the values estimated by the model and bootstrap resampling. We have therefore chosen to add the parameters estimated by the Jackknife in table 2.
- For Monte Carlo simulation, why did you decide to simulate the drug level on Day7?
There is no recommendation for the use of TDM when using isavuconazole. In our case, the half-life of isavuconazole varies between 90 and 150 hours. Theoretically, 50% of steady state is reached after one week of treatment. Combined with a loading dose, this leads to a near steady state. When analyzing the results of the simulation, only the non-dialysis patients did not fully reach steady state at the high dose. In the context of clinical use, it is relevant to monitor treatment with a sample at D7.
- How long of RRT period of most patients?
The average duration of dialysis was 24 days (min - max: 8 - 41 days).
And RRT was initiated before or after starting treatment?
Of the 6 patients who were dialyzed during treatment, 50% were dialyzed prior to treatment initiation. This was adjusted for by tagging as regressor the RRT covariate on dialysis and non-dialysis time.
We precised it in Table S1 for RRT initiation before or after treatment and added a complement in the Results section (line 241-243):
To account for the change in the RRT covariate over time (dialysis or non-dialysis periods), it was tagged as a regressor.
, and these simulated regimens are made for the patients on RRT who are subsequently infected with CAPA?
Yes, simulations were done for patients with Covid-19 complicated by invasive aspergillosis (CAPA), on RRT or not.
If COVID-19 patients were infected CAPA and already started isavuconazole before RRT, can you recommend using your simulation to guide the dosing?
Yes, we also recommend to use simulation with status CAPA non-RRT.
Requiring Re-loading?
In the conclusion section, we have made a recommendation along the following lines (449 – 460):
In this particular population where cumulative supportive care seems to have an impact on the PK of isavuconazole, this confirms the need for TDM in CAPA patients, especially with those under RRT. For CAPA patients (RRT or non-RRT), in order to secure and optimize therapeutic management, 800 mg/day during 3 days followed by 400 mg/day seems necessary. At this stage, two approaches are possible: either the empirical approach consisting of performing TDM at the end of the loading dose and readjusting the maintenance dose later, or the rational approach consisting of taking a blood sample at H48 in order to adjust the maintenance dosage at H72 using the PK model. In the event that the patient is dialyzed several days after the initiation of isavuconazole, TDM should be performed to assess the impact of dialysis on the patient's exposure. In case of underdosing, a re-loading dose should be necessary.
- Line212: It should be better not to use “therapeutic target 2-5 mg/L”, because now we don’t know the therapeutic range of isavuconazole. You should change to another wording.
It is difficult to know which term to use to define a treatment zone. We already described in the introduction section that the lower limit of efficacy was 1 or 2 mg/L, depending on the study. Therefore, we have replaced the term "target" with "range", which has less connotations. Historically, this is the SWAB (Dutch Working Party on Antibiotic Therapy). SWAB guidelines for the treatment of invasive fungal infections. Revised version. Published: 14 December 2017), which defined a therapeutic range of 2-4 mg/L. Later, the study by Furfaro et al (https://doi.org/10.1093/jac/dkz188) defined a toxicity threshold of 5 mg/L, in particular to describe the occurrence of gastrointestinal adverse drug reactions. Furthermore, the clinical breakpoint for Aspergillus fumigatus/flavus is 2 mg/L (EUCAST), so this therapeutic range seems to be emerging: 2 - 5 mg/L. Although there is still uncertainty about the therapeutic range of isavuconazole, many consider that the therapeutic range is between 2 and 5 mg/L. (https://doi.org/10.3390/antibiotics10050487).
We added a paragraph concerning therapeutic range in the discussion section (line 338 - 344): Microbiologically, the EUCAST threshold in 01/2022 was set at 2 mg/L for isavuconazole [28]. A plasma concentration ≥ 2 mg/L seems to provide better exposure for all patients [48]. The study by Furfaro et al. [20] defined a toxicity threshold of 5 mg/L, in particular to describe the occurrence of gastrointestinal adverse drug reactions. Although there is still uncertainty about the therapeutic range of isavuconazole, many consider that the therapeutic range is between 2 and 5 mg/L [21].
- Any covariate affecting on the volume of distribution? Such as serum albumin at each time point, because most of patients were hypoalbuminemia and isavuconazole is highly bound to plasma protein.
There were no covariates that affected volume of distribution. Although isavuconazole is highly bound to albumin, hypoalbuminemia is observed in our cohort. It could have an effect on Vd, but the patient cohort is relatively homogeneous and the number of patients is small, making it difficult to show the effect of a covariate such as hypoalbuminemia on Vd.
We added this in the discussion section (line 395 - 402):
Finally, for the volume of distribution, we found a mean value of 850 L, very close to that reported by Cojutti et al. (835 L) [55], but much higher than those reported by Desai et al. [56], Kovanda et al. [57], or Wu et al. [58]. This elevated volume of distribution could be explained by an elevated BMI in our cohort (29.2 kg/m²). Moreover, hypoalbuminemia was observed in our patients, which could also have played a role in the increase of the volume of distribution but the small number of patients in our cohort with all hypoalbuminemic patients does not allow sufficient statistical power to demonstrate the effect of this covariate on this PK parameter.
- You should state that for non-RRT patients, there was not any covariate affecting on isavuconazole PK parameters. Because non-RRT patients were the most of all patients.
We have already made this factually clear in the results (line “RRT had a significant effect on clearance (Cl), and was the only covariate retained in the final model”). Discussing it when it is already said that we lack power might be redundant.
- Discussion section, Line323 ”We developed a model to reliably predict plasma isavuconazole concentrations in our population for up to 50 days of treatment to capture the steady state of isavuconazole.” I knew that you followed the patients up to 50 days of treatment, but I’m not sure that you can use this sentence to claim that your model is reliably predicting isavuconazole for up to 50 days of treatment. Your model could predict well during the first two weeks of treatment according to your most available blood concentration.
We totally agree, as the number of concentration points between 21 and 50 days is not in the majority, it seems presumptuous to formulate things in this way. We have changed the sentence (line 310 – 311):
We developed a model to reliably predict plasma isavuconazole concentrations in our population for up to 21 days of treatment to capture the steady state of isavuconazole (as most available blood concentrations are collected between 4 – 21 days).
- It should be better if you can discuss more about the physicochemical properties of isavuconazole and the mechanism of each RRT mode to remove isavuconazole, such as isavuconazole MW around 437 dalton can be removed by CVVHD efficiently to relate to your PPK model result, to highlight this concern to the readers.
For more clarity, we added a paragraph in the discussion section line (361 – 375):
Molecular weight of isavuconazole is less than 500 daltons, but its very lipophilic character (logP = 3.5), its high protein binding (predominantly albumin, > 99%), and its high volume of distribution > 1L/kg do not allow us to say that isavuconazole is a good candidate for conventional hemodialysis. However, in critical ill circumstances with severe hypoalbuminemia, the binding of isavuconazole to other plasma components, and pH abnormalities, all these biological variations make it difficult to predict the dialyzability of isavuconazole. In a study of neutropenic ICU patients, isavuconazole plasma concentrations were significantly lower after SLED therapy (3.36 vs. 5.73 mg/L; P < 0.001), representing a 42% decrease in drug levels [29]. In our study, patients with lower plasma concentrations were also SLED patients. The characteristics of the dialysis could be a factor in lowering plasma concentrations, but our small number of patients does not allow us to determine the effect of dialysis mode, filter type, blood flow rate, and effluent flow rate, but to consider only whether the patient is on dialysis or not (Table S1).
Minor comments
- Line77, “In a recent post-hoc analysis of the SECURE trial, Desai et al. found no relationship between exposure as represented by the area under the MIC curve and clinical outcome.” Question: I’m not sure that these bold letters are correct? You mean AUC/MIC? Or area under concentration time curve over MIC.
This is a transcription error, as the authors were only talking about the AUC. Thank you for pointing this out. This has been corrected in the text.
- In Table 1, I’m not familiar with the unit of white blood cells and platelet count, G/L = gram/litre?.
It is G/L = Giga/L
